



# Towards a more complete quantification of the global carbon cycle

Miko U.F. Kirschbaum[1], Guang Zeng[2], Fabiano Ximenes[3], Donna L. Giltrap[1], John R. Zeldis[4]

[1] Landcare Research – Manaaki Whenua, Private Bag 11052, Palmerston North, New Zealand
[2] National Institute of Water & Atmospheric Research, Private Bag 14901, Wellington 6021, New Zealand
[3] Forest Science Unit, New South Wales Department of Primary Industries, Locked Bag 5123, Parramatta NSW 2150, Australia
[4] National Institute of Water & Atmospheric Research, PO Box 8602, Christchurch 8011, New Zealand

*Correspondence to*: Miko U.F. Kirschbaum (KirschbaumM@LandcareResearch.co.nz)

**Abstract.**

The main components of global carbon budget calculations are the emissions from burning fossil fuels, cement production, and net land-use change, partly balanced by ocean $CO_2$ uptake and $CO_2$ increase in the atmosphere. The

remaining difference between these terms is referred to as the residual sink, assumed to correspond to increasing carbon storage in the terrestrial biosphere ($\Delta B$). It is often used to constrain carbon exchange in global earth-system models. More broadly, it guides expectations of autonomous changes in global carbon stocks in response to climatic changes, including increasing $CO_2$, that may add to, or subtract from, anthropogenic $CO_2$ emissions.

However, a budget with only these terms omits some important additional fluxes that are important for correctly

inferring $\Delta B$. They are cement carbonation and fluxes into increasing pools of plastic, bitumen, harvested-wood products, and landfill deposition after disposal of these products, and carbon fluxes to the oceans via wind erosion and non-$CO_2$ fluxes of the intermediate break-down products of methane and other volatile organic compounds. While the global budget includes river transport of dissolved inorganic carbon it omits river transport of dissolved and particulate organic carbon, and the deposition of carbon in inland water bodies.

Each one of these terms is relatively small, but together they can constitute important additional fluxes that would significantly reduce the size of the inferred $\Delta B$. We estimate here that inclusion of these fluxes would reduce $\Delta B$ from the currently reported 3.6 down to only about 2.1 GtC yr$^{-1}$ (excluding losses from land-use change). The implicit reduction in the size of $\Delta B$ has important implications for the inferred magnitude of current-day biospheric net carbon uptake and the consequent potential of future biospheric feedbacks to amplify or negate net anthropogenic $CO_2$

emissions.

## 1. Introduction

In its summarised form, the global carbon cycle is usually expressed in the form of six main fluxes (Le Quere et al., 2018; Figure 1). Carbon is added to the atmosphere by the burning of fossil fuels (9.0 GtC yr$^{-1}$), cement production

(0.4 GtC yr$^{-1}$) and ongoing deforestation, mainly in the tropics (1.3 GtC yr$^{-1}$). Some fossil fuels (0.4 GtC yr$^{-1}$) are also utilised for the manufacture of other products, like plastics, or are incompletely combusted and thus do not directly emit $CO_2$ to the atmosphere. The atmospheric $CO_2$ concentration has increased to over 400 ppm through annual additions of about 4.7 GtC yr$^{-1}$ whereas the oceans are still close to their pre-industrial effective equilibrium



concentration of 280 ppm. This difference constitutes a driving force for ocean $CO_2$ uptake, estimated at 2.4 GtC $yr^{-1}$ (Le Quere et al., 2018).

Summing these various fluxes results in an imbalance of 3.6 GtC $yr^{-1}$, often referred to as the 'residual sink'. This flux cannot be directly and independently estimated, but is derived as the residual remaining after estimation of the other terms. In the most recent budget, this has been separated into an estimated 'land sink', based on earth-system models, and a remaining 'budget imbalance' (Le Quere et al., 2018).

The size of the residual sink is often implicitly or explicitly equated with carbon uptake by the terrestrial biosphere (e.g. Ciais et al., 2013; Sitch et al., 2015; Arneth et al., 2017; Huntzinger et al., 2017). A sink of 3.6 GtC $yr^{-1}$ suggests that one third of anthropogenic emissions might be balanced by biospheric carbon uptake and storage. The size of this flux is even more important for future trends in biospheric uptake that could provide an important positive or negative feedback for atmospheric $CO_2$ changes (Cramer et al., 2001; Jones et al., 2013). If the magnitude of terrestrial uptake is over- or underestimated, it would lead to incorrect inference about the strength of future feedback processes from the terrestrial biosphere on the earth's net carbon budget.

However, in the global carbon budget as presented in Figure 1, several important fluxes have been omitted. In the present work, we aim to provide a quantification of these additional terms based on values found in the existing literature or derived in the current work, and thereby more completely quantify the global carbon cycle. In addition, we estimate the increase in carbon stored in the terrestrial biosphere, $\Delta B_{incLUC}$ by explicitly accounting for the carbon flux into additional carbon-storage pools or through pathways not previously included in global budget calculations. We also estimate the net change in carbon stored in the terrestrial biosphere, $\Delta B$, to refer to the change in stored carbon excluding the effects of land-use change.

Hence, the present works aims to quantify these additional terms:

1) Net increases in the pools of harvested-wood products, plastic, bitumen, rubber, leather and textiles while they are in service;
2) Net increases of carbon in anaerobic landfills after subsequent disposal of these products;
3) The carbonation of previously manufactured cement products;
4) River transport from the land to the oceans as dissolved or particulate organic carbon;
5) Carbon deposition in inland water bodies;
6) Transfer of carbon from the land to oceans via aeolian transport either attached to mineral dust or as charcoal;
7) Fluxes of non-$CO_2$ gases, principally methane and NMVOCs (non-methane volatile organic compounds) and their intermediate break-down products.

Of these, $CO_2$ fluxes associated with cement carbonation (Xi et al., 2016), and carbon deposition in fresh-water bodies (e.g. Regnier et al., 2013) constitute obvious fluxes from the atmosphere into relevant storage pools that have not been included in the global budgets. There is also a sizeable net flux into the pool of harvested-wood products (Lauk et al., 2012). This flux has already been included in net land-use change calculations (Le Quere et al., 2018), but in the interest of transparency, it would be preferable if that flux were quantified more explicitly.

Net carbon fluxes into the pools of plastic and bitumen and subsequently into anaerobic landfills have also been included indirectly by accounting for only an assumed fraction of fossil-fuel carbon being oxidised (e.g. Marland and Rotty, 1984; Le Quere et al., 2018). A small fraction of fossil fuels is used for manufacturing products, such as plastic and bitumen, and of the fossil fuels that are burnt, another small fraction is only incompletely combusted leading to less than 100% being converted to $CO_2$ (Marland and Rotty, 1984). Based on these considerations, Marland and Rotty (1984) estimated oxidation fractions of 98%, 91.8% and 98.2% for the utilisation of gas, liquid and solid fuels, respectively. These terms are then applied to fossil-fuel production data to derive fossil-fuel-based $CO_2$ emission rates





(e.g. Andres et al., 2012). In the interests of greater transparency, it would be desirable, however, if fluxes through these key product pathways were more explicitly accounted and reported in future global emission budgets.

Carbon transport to the oceans through river transport (Regnier et al., 2013), aeolian fluxes (e.g. Romankevich et al., 2009) or gas fluxes by carbon compounds other than $CO_2$ all constitute additional carbon fluxes from the land to the
oceans. These fluxes are only incompletely accounted for in the standard quantification of the global carbon cycle, and a more complete quantification is given below. The significance of the different terms in land-ocean exchange are discussed in the next section.

### 2. Ocean Exchange

In deriving the global carbon budget, Le Quere et al., (2018) used estimates of air–ocean $CO_2$ exchange rates ($T_{ia}$ in
Fig. 2) and added the transport of inorganic carbon via river transport ($T_{ir}$, Fig. 2) with the aim of describing the anthropogenic carbon budget (Jacobson et al., 2007; Le Quere et al., 2018). However, this omits other important transport pathways as illustrated in Figure 2. The ultimate key fluxes are the net transport of carbon from the shallow to the deep ocean (or to ocean-floor deposition in shallow seas) as either inorganic $CO_2$ ($H_2CO_3$, $HCO_3^-$, $CO_3^{2-}$), including $CaCO_3$ in solid form, or in any soluble or particulate organic form. Hence, the relevant total carbon transfer,
$T_c$, can be described as $T_c = T_{id} + T_{od}$ where $T_{id}$ and $T_{od}$ are the net carbon transfers to the deep ocean of inorganic and organic carbon, respectively. The shallow ocean is too small for significant carbon storage, but the deep ocean has a huge carbon-storage capacity. The shallow ocean is important, however, as the interface between the ocean and the atmosphere and where $pCO_2$ measurements are taken for the estimation of net $CO_2$ exchange.

In the ocean, organic and inorganic forms of carbon continuously interchange. Inorganic carbon is fixed and converted
into organic forms by photosynthetic organisms. As these organisms are eaten by larger organisms, carbon is respired in inorganic form. The sizes of these conversion fluxes are not important in the present context, as carbon can ultimately be transferred to depth in either organic or inorganic form. The net flux of inorganic carbon from the deep ocean may even be negative (outgassing), with net carbon transfer to depth reliant on organic carbon transfer.

As transfers $T_{id}$ and $T_{od}$ are difficult to measure directly, the flux $T_c$ is normally approximated as $T_c = T_{ia} + T_{ir}$ while
the fluxes of organic carbon from atmospheric transfer or river transport, $T_{oa}$ and $T_{or}$, are ignored and omitted from the estimated global fluxes. Instead, we propose that the more appropriate total flux should be calculated as: $T_c = T_{ia} + T_{ir} + T_{oa} + T_{or}$. Below, we quantify the different fluxes of organic carbon to the oceans to complete the overall sums.

### 3. Calculation Details

For comparison between the residual sink and estimates of carbon exchange of the land biosphere, we used the data
given by Le Quere et al. (2018) as land sink and budget imbalance for different years. Previous carbon budgets (e.g. Le Quere et al., 2016) provided numbers denoted as residual sink activity. In the latest budget, this has been disaggregated into a land sink, estimated from biosphere models, and a budget imbalance term (e.g. Le Quere et al., 2018). The sum of these two terms equates to the previously given residual sink, R.

Changes in the terrestrial C stock were calculated as:

$\Delta B = R - R_d - R_p - R_i - D - V - C - P - B - L + N$                       (1)



$$\Delta B_{incLUC} = \Delta B + LUC \qquad\qquad (2)$$

where $R_d$ is river transport as dissolved organic carbon, $R_p$ is river transport as particulate organic carbon, $R_i$ is carbon deposition in inland water ways, D is carbon transport to the oceans as aeolian dust deposits, V is transfer from volatile intermediate oxidation products of methane and NMVOCs, C is carbon storage in cement carbonation, P, B and L are the changes in carbon stored in plastics, bitumen and landfills, respectively, and N is the non-oxidised fraction of fossil consumption that has been implicitly included in previous budgets. The terms P, B and L therefore largely cancel out the term N, but the calculations are made more explicit here.

The term $\Delta B_{incLUC}$ refers to the actual change in the total terrestrial biosphere, including changes due to land-use change, and $\Delta B$ refers to biospheric carbon-stock changes due to physiological and age-class effects, but excluding land-use change. LUC is the carbon-stock change due to land-use change with negative numbers denoting net losses to the atmosphere. Of these various components, no temporal trends were available for $R_o$, $R_p$, $R_i$, D and V, but temporal patterns could be included for P, B, C, L and N based on the work of Lauk et al. (2012) and Xi et al. (2016) and calculated, following Marland and Rotty (1984), as:

$$N = F (0.02\ g + 0.082\ 1 + 0.018\ s) \qquad\qquad (3)$$

where F is total fossil-fuel consumption and g, l and s are the percentages of gas, liquid and solid fuel in the global mix of fossil fuels, estimated as constant percentages of 17.0%, 41.8% and 41.2% since 1959, and the constants in eq. 3 have been taken from Marland and Rotty (1984).

## 4. Wood products, plastics, bitumen, cement carbonation

For harvested-wood products, plastic and bitumen in service by human societies, the relevant quantity in the present context is the net increase in the size of these pools. At the end of their service lives, plastic and harvested-wood products, especially paper products, may be re-used, recycled, or disposed of either by incineration or disposal in landfills. If they are incinerated in waste-to-energy facilities, $CO_2$ is released to the atmosphere immediately, and if they are re-used or recycled, the products re-enter the 'in-service' pool. Alternatively, these products may be deposited in landfills in countries that use landfills as part of their waste management strategies, which will be discussed in the next Section.

For harvested-wood products in service, net increases in carbon stocks primarily correspond to the pool of long-lived structural wood products, such as housing frames. Paper products, on the other hand, tend to have short service lives and do not build up to sizeable pools even though fluxes through these pools can be substantial. This can include multiple passes through the active-service pool because paper products may be recycled repeatedly before eventual disposal. Le Quere et al. (2018) included a simple term in the calculations of net land-use change that accounted for harvested-wood products. They assumed that a fraction of the wood lost through land-use change was not directly lost as $CO_2$ to the atmosphere but retained in harvested-wood products. However, we believe that a more explicit representation of this pool would be desirable for greater transparency.

In the case of cement carbonation, the flux is associated with the degeneration of previously manufactured cement. Cement manufacture is essentially the calcination of $CaCO_3$ into $CaO$ under high temperature. The resultant $CO_2$ release is included in global carbon budgets (Andrew, 2018) and accounts for about 4% of total anthropogenic $CO_2$ emissions (Le Quere et al., 2018). When cement is subsequently exposed to rain and natural $CO_2$ concentrations, the



process is reversed, and $CO_2$ is reabsorbed, replacing oxygen bound to calcium (Xi et al., 2016). This causes the gradual degradation of cement, with the rate of degradation essentially determined by the slow diffusion of $CO_2$ into any cement products.

Essentially all cement is subject to that kind of degradation, with its rate decreasing with the thickness of the cement layer. Thinner layers of mortar therefore degrade faster than more solid concrete structures. When a building is demolished, cement carbonation tends to increase as cement becomes fragmented, thereby opening new surfaces that allows more rapid diffusional penetration of $CO_2$. The rate of cement carbonation can, therefore, be approximated as being proportional to total cumulative past cement production. Hence, global carbonation rates were likely to have been low in the 1950s, then increased gradually to the 1990s (Fig. 3a), with much more substantial increases since then. Using statistics of historical cement production in different categories, Xi et al. (2016) estimated recent uptake rates of about 250 MtC yr$^{-1}$, with uptake rates expected to continue increasing into the future.

The socio-economic models of Kayo et al. (2015) and Brunet-Navarro et al. (2016) have shown that in poorer societies, wood use per person increases with increasing wealth (quantified as gross domestic product, GDP, per capita, cp$^{-1}$). However, that relationship saturates at intermediate values of GDP cp$^{-1}$ and even becomes negative for the wealthiest societies. Lauk et al. (2012) estimated that humans own on average approximately 1 tC cp$^{-1}$ in harvested-wood products. If that figure remained constant, one could assume an annual increase of the global pool by about 80 MtC yr$^{-1}$ purely driven by global population growth. If wood use per person is also increasing, as shown by Kayo et al. (2015), it would result in an increase in the global harvested-wood-products pool by more than 80 MtC yr$^{-1}$. Winjum et al. (1998) and Lauk et al. (2012) estimated changes in the harvested-wood products pool from analysis of wood-production statistics and assumption about product longevities. Winjum et al. (1998) estimated an annual increase of about 140 MtC yr$^{-1}$, while Lauk et al. (2012) provided a slightly smaller estimate of recent increases by just under 100 MtC yr$^{-1}$. They also provided historical estimates over the 20[th] century (Fig. 3a).

However, for most countries and wood-product categories (paper, wood panels, and sawn wood), there are no reliable service life factors. Global analyses therefore have had to rely on the use of generic factors, such as IPCC default Tier 2 half-lives (IPCC 2014). Lauk et al. (2012) considered the need to use these generic factors as the primary cause of the large uncertainties in their estimated carbon fluxes into harvested-wood-product pools. Lauk et al. (2012) also estimated fluxes into the pools of bitumen, used mainly for road construction, and plastics (Fig. 3a). Fluxes started from very low values before 1950 but have increased steadily and are now similar to fluxes into the pool of harvested-wood products.

The combined flux from these four fluxes was estimated to have been less than 50 MtC yr$^{-1}$ in 1950, but increased steadily to about 300 MtC yr$^{-1}$ by 2000 (Fig. 3b). The rate of uptake has increased more sharply since then, driven mainly by increasing cement carbonation, and is estimated to have reached about 450 MtC yr$^{-1}$ by 2010 (Fig. 3b).

## 5. Landfill Storage

At the end of their service lives, products may be disposed of in landfills, where conditions may be aerobic, semi-aerobic or anaerobic depending on their management (IPCC 2006). If materials are kept under anaerobic conditions, their effective storage life can be extended substantially, with very slow decomposition and resultant carbon loss (Wang et al., 2011, 2015; Ximenes et al., 2015, 2018).



Wood and plastics are particularly persistent after disposal unless they are incinerated. Bitumen is not usually disposed of, but if roads are renewed, old bitumen is typically recycled, with only minor losses (Lauk et al., 2012). Textiles, rubber and leather make additional minor contributions to total landfill carbon stocks. With all categories added together, anaerobic landfills can thus store large amounts of carbon.

Lauk et al. (2012) estimated total annual disposal rates of various key products (Fig. 4), estimated at nearly 500 MtC yr$^{-1}$. While Figure 4 clearly shows the historical pattern of product disposal, it does not indicate what quantities of products are disposed of in anaerobic landfills. To the best of our knowledge, there have been no prior estimates of global net carbon stock changes in landfills. We have therefore attempted to provide a first global estimate of waste disposal in anaerobic landfills and consequent annual changes in landfill carbon stocks (Table 1).

Accounting for annual landfill fluxes of different waste streams, their dry-matter percentages, carbon contents and relative permanence under anaerobic conditions, we estimated changes in long-term carbon pools in landfills for different product categories. The temporal pattern of breakdown in landfills is not clear. One normally describes the breakdown of products as an exponential decay process which can be described with a simple decay constant or its inverse, the residence time. However, under anaerobic conditions, breakdown effectively ceases completely, and a

permanence factor essentially separates products into a fraction that breaks down over a relatively short time frame and a second fraction that never breaks down. The sizes of these fractions are determined by their associated degradability, such as cellulose to lignin ratios, and the biophysical conditions within landfill sites (e.g. Barlaz, 2006).

Paper and paperboard constituted the largest disposal category, but because of its relatively fast rate of degradation (Wang et al., 2011, 2015; Ximenes et al., 2015, 2018), its contribution to increasing carbon stocks is only minor.

Although less wood and engineered wood products (e.g. plywood, particleboard) are disposed of in landfills than of paper and paperboard, it leads to a higher estimated storage flux because wood is highly resistance to degradation under anaerobic conditions. Plastics have the highest storage flux (42 MtC yr$^{-1}$) because of their high disposal rate, high carbon content and very high persistence.

Using the detailed data and assumptions in Table 1, we calculated a net carbon change in landfill storage by 88 MtC

yr$^{-1}$.

## 6. River Transport

A large amount of carbon is transported from the terrestrial biosphere to the oceans through river flow. Carbon can be transported as dissolved inorganic (DIC), dissolved organic (DOC) or particulate organic (POC) carbon (Ward et al., 2017). These fluxes are difficult to quantify because of the enormous diversity of river systems (Regnier et al., 2013;

Mendonca et al., 2017), and the large episodic contribution to some fluxes, especially of particulate organic carbon, by infrequent flood events. Net fluxes into and out of inland water systems also consist of multiple entry points and large outgassing as some organic materials are broken down and respired as $CO_2$ before they can be deposited in lake sediments or the oceans while simultaneously, some new carbon is fixed through aquatic photosynthesis.

Mendonca et al. (2017) documented the largest reported emissions rates for small reservoirs, with variability that

extended over three orders of magnitude, yet global estimates had to be based on a mere 59 available point estimates. The combined surface area of these smaller reservoirs is fortunately much smaller than that of large lakes which reduce the importance of that uncertainty. Larger lakes had similar relative variabilities in observed rates but smaller averages.



However, the small number of available observations clearly prevents the size of this globally important flux to be estimated with high confidence (e.g. Regnier et al., 2013).

Despite these difficulties, various authors have attempted to provide global estimates of the key fluxes (Table 2; Fig. 5). Most authors have estimated total influx to inland water ways as between 2700 and 2900 MtC yr$^{-1}$, while the recent

work by Drake et al. (2018) gave a much larger estimate of 5100 MtC yr$^{-1}$ (Table 2). Of that amount of carbon entering inland water ways, different authors have estimated outgassing losses between 750 and 2120 MtC yr$^{-1}$, with the estimate of Drake et al. (2018) again being much larger at 3900 MtC yr$^{-1}$. If one uses these estimates, together with some extra inputs mineral weathering, this leaves about 1500 MtC yr$^{-1}$ to be either deposited in inland water bodies or transported to the oceans (Table 2).

Apart from the older work of Cole et al. (2009), most other authors estimated total inland deposition as 600 MtC yr$^{-1}$ and total flux to the ocean as 900 MtC yr$^{-1}$. This is broken down into estimated dissolved inorganic carbon flux of 450 MtC yr$^{-1}$, particulate organic flux of about 250 MtC yr$^{-1}$ and dissolved organic carbon of 200 MtC yr$^{-1}$. Romankevich et al. (2009) estimated an additional contribution of 47 MtC yr$^{-1}$ from coastal erosion, ground-water influx and glacial run-off.

Considering the evidence used by the various authors, we consider total carbon flux to inland waterways to most likely be about 2900 MtC yr$^{-1}$ (Fig. 5; Table 2). About half of that carbon (1400 MtC yr$^{-1}$) is lost from water ways by outgassing, although neither of those estimates are needed for explicit inclusion in the global budget.

The important fluxes are the transport to the oceans, estimated to be about 900 MtC yr$^{-1}$ and consisting of 450 MtC yr$^{-1}$ DIC, 200 MtC yr$^{-1}$ DOC and 250 MtC yr$^{-1}$ POC, with general convergence between different studies (Table 2).

The DIC flux is already included in the estimate of total inorganic ocean uptake, but the DOC and POC fluxes have not been included in the global summary numbers of Le Quere et al. (2018).

In addition, between 60 and 250 MtC yr$^{-1}$ are deposited in lakes and water reservoirs (Mendonca et al., 2017). Other studies have also included deposition in wetlands, floodplains and sediments for total deposition estimated to be about 600 MtC yr$^{-1}$ in all inland water bodies (Tranvik et al., 2009; Aufdenkampe et al., 2011). This flux has also not yet

been included in the global flux quantification of of Le Quere et al. (2018).

### 7. Aeolian fluxes

Carbon can also be transported from the land to the oceans by aeolian transport through wind erosion of dust particles (Zender et al., 2003; Webb et al., 2012). These carbon fluxes to the ocean are not captured in air–sea $CO_2$ exchange but add to the total flux of carbon from the land to the ocean (see Fig. 2).

Romankevich (1984) estimated aeolian carbon flux as 320 MtC yr$^{-1}$, while Romankevich et al. (2009) estimated it as 96 MtC yr$^{-1}$. Estimates can also be based on independently estimating the annual flux of aeolian dust and their carbon concentrations. Mahowald et al. (2005) summarised the different available estimates of the total aeolian dust flux as 1500–2000 Mt(dust) yr$^{-1}$. Assuming source carbon concentrations between 1 and 2% (Webb et al., 2012; Chappell et al., 2013) and a 2.5-fold enrichment of carbon concentrations in dust relative to source concentrations (Webb et al.,

2005), it leads to a global flux estimate of 50-100 MtC yr$^{-1}$.





### 8. Charcoal

A sizable fraction of annually produced biomass is burnt each year (Kuhlbusch and Crutzen, 1995). Savannah vegetation is particularly prone to annual burning, and a fraction of burnt material is not combusted completely but remains as charcoal, estimated as 50–270 MtC yr$^{-1}$ (Forbes et al., 2006). A small fraction of that will become airborne, either during fires themselves or in subsequent wind storms, and a small proportion of that airborne fraction will be transported to the oceans. Forbes et al. (2006) estimated this flux to be only small at less than 10 MtC yr$^{-1}$.

### 9. Methane and NMVOCs

The principal gas transfer of carbon to the oceans is via $CO_2$, but carbon can also reach the ocean in organic gaseous form (Fowler et al., 2009). The annual combined flux of methane and non-methane volatile organic compounds (NMVOCs) is estimated to be about 1.3 GtC yr$^{-1}$, with methane fluxes contributing about 500 MtC yr$^{-1}$ (Ciais et al., 2013) and NMVOCs, about 800 MtC yr$^{-1}$ (Fowler et al., 2009), more than half of which is isoprene. Most of these compounds are oxidised in the troposphere, with methanol, methyl hydroperoxide and formaldehyde as key intermediate oxidation products (Fig. 6). If these compounds were fully oxidised to $CO_2$ in the atmosphere, there would be a simple closed loop between production by the terrestrial biosphere and atmospheric oxidation, but any transfers to the ocean by compounds other than $CO_2$ constitutes an additional carbon transfer from land to the ocean (see Fig. 2) that is not otherwise captured in the budget.

This transfer can be by direct transfer to the surface ocean or after prior solution in raindrops. This direct flux of methane and isoprene is probably small due to their low water solubility. However, under partial oxidation in the atmosphere, major intermediate products are methanol, organic acids, and formaldehyde, which are all highly soluble in water and can be deposited in the oceans as wet (after dissolution in rain or fog) or 'dry' deposition when gases dissolve directly in ocean water. As we are not aware of prior estimates of this flux, we have estimated wet and dry deposition of the relevant compounds here (Table 3). Details of the calculation methods are given in Supplemental Information.

The compounds in Table 3 show the quantitatively important intermediate oxidation products of methane, isoprene, and other NMVOCs. We calculated total ocean dry deposition of 10.8 MtC yr$^{-1}$ and wet deposition of 39.1 MtC yr$^{-1}$, which together account for around 27% of total surface deposition (with 73% assumed to occur over land). Some of these intermediate products have short lifetimes and are therefore mainly deposited close to their point of production, which is mostly over land areas.

Summing these various fluxes provides an additional ~50 MtC yr$^{-1}$ of non-$CO_2$ flux from the atmosphere to the oceans. Any estimation of global fluxes depends strongly on deposition schemes, chemical mechanisms, and terrestrial NMVOC emissions, which vary among global models and are poorly constrained by observations. Hence, there are considerable uncertainties in these estimated fluxes, as demonstrated by Jacob et al. (2005), for example, in the case of the global methanol budget. They summarised the results of various previous studies and reported global dry deposition on the oceans estimated by different models of 0.3–50 Mt(CH$_3$OH) yr$^{-1}$ plus total global wet deposition of 9–50 Mt(CH$_3$OH) yr$^{-1}$ without separation between land and ocean deposition.

This illustrates the remaining level of uncertainties in these global estimates. There are also considerable differences in isoprene and monoterpene oxidation mechanisms among the models, in particular the formation of intermediate



products from isoprene oxidation (e.g. Paulot et al., 2009). Some further information on these uncertainties is given in on-line Supplemental Information.

## 10. Summary of the Main Fluxes in the Global Carbon Cycle

Consideration of these additional pools and fluxes reduces the estimated additional carbon stored in the terrestrial biosphere, ΔB, from 3.6 to 1.9 GtC yr⁻¹ (Fig. 7, Table 4). While none of the various extra fluxes are particularly large or important on their own, added together, they reduced the size of the inferred terrestrial biosphere sink by about 1.5 GtC yr⁻¹.

For greater transparency, it would also be desirable to explicitly include harvested-wood products and landfill pools. The associated carbon flux is already included under the net-land-use calculations (Le Quere et al., 2018). Inclusion of a harvested-wood-products pool, therefore, would not affect the size of the residual sink, but it would require a corresponding adjustment of the net land-use-change flux.

The fluxes into increasing pools of plastics, bitumen, and waste storage in landfills are clear and obvious fluxes that are quantitatively important and additional to fluxes currently considered by Le Quere et al. (2018). Their effect on the overall budget had, however, also already been included indirectly in the fossil-fuel fluxes through a term that accounts for incomplete oxidation of fossil fuel use (Marland and Rotty, 1984). The fluxes into the increasing pools of plastic and bitumen are reasonably well constrained. The flux into increasing landfill carbon storage is less well constrained, as we could find no prior global assessment of this flux. We have provided a first such global estimate in the present work, but significant uncertainty remains due to incomplete knowledge of regional details of the key properties of different waste streams. In any case, explicit inclusion of fluxes into these storage pools would be desirable to increase transparency of the overall global carbon budget.

These incomplete oxidation terms for fossil-fuel use (Marland and Rotty, 1984) account for incomplete combustion during energy generation and for non-fuel uses. That has been represented explicitly in Figure 7. For internal consistency, the fossil-fuel consumption rates have therefore been increased by 0.4 GtC yr⁻¹ so that non-fuel uses are given explicitly in Figure 7. While for transparency, it would be desirable to make these fluxes explicit, it would not affect the estimated size of the residual sink.

Cement carbonation is an additional sink that is likely to increase in proportion to the cumulative total amount of manufactured cement and is therefore likely to increase further into the future. Its magnitude is also reasonably well constrained and is clearly bounded by the total historical cement production. This flux has so far been omitted from the global carbon budget, and its inclusion reduces the size of the residual sink.

River transport as dissolved and particulate organic carbon is also reasonably well constrained. However, the enormous heterogeneity of river types makes confident assessment difficult. This is further compounded by the disproportionate importance of rare flooding events that can episodically transport large quantities of particulate organic matter. Nonetheless, the various global estimates are converging on similar flux estimates (e.g. Regnier et al., 2013; Drake et al., 2018).

A fraction of this organic carbon flux is oxidised in the shallow ocean, leading to outgassing in some regions (e.g. Borges et al., 2005; Jacobson et al., 2007). Another fraction is transferred to the ocean floor or the deep ocean in organic form. Particulate organic carbon associated with soil minerals is particularly prone to direct sinking to the



ocean floor. That mineral-associated fraction should obviously be included. The fraction that is oxidised in the shallow ocean and converted to inorganic carbon will increase the surface $pCO_2$ (partial pressure of $CO_2$). This lowers the atmosphere to ocean $CO_2$ gradient and reduces ocean $CO_2$ uptake, or can even lead to outgassing. Calculations of ocean $CO_2$ uptake by gaseous exchange should correctly reflect that, but total transfer of $CO_2$ to the surface ocean will

be the combined flux of air–sea exchange plus the additional contribution of organic carbon that found its way to the ocean by aeolian or river transfer or by gas transfer of non-$CO_2$ carbon compounds. Regardless of those further transformations, Figure 2 showed that it would be appropriate to include this flux of organic carbon as an important addition to the overall budget.

Deposition of carbon in inland water ways is another quantitatively important flux into an additional carbon storage
pool that should be included in the overall budget. With the increasing regulation of water ways and the construction of more dams on the world's rivers (e.g. Regnier et al., 2013), and possible increases in erosion fluxes (e.g. Yang et al., 2003), this flux is also likely to continue to increase into the future

Some of the erosion-related component of this flux constitutes a simple lateral carbon transfer from erosion sites to some downstream water way with no net effect on the atmosphere. However, most denuded erosion sites can
eventually regain their lost soil organic carbon. While that process is slow and may remain incomplete, the resultant potential carbon gain needs to also be factored in (van Oost et al., 2007). It would, therefore, be too simplistic to ignore inland deposition as just a lateral transfer. In its totality, erosion may act as a net sink or source of carbon to the atmosphere. For global carbon accounting purposes, it means that inland deposition should be included, but any changes in soil carbon stocks also need to be quantified to complete the overall balance.

The next relevant flux is the transport of carbon attached to aeolian dust or charcoal. Again, this flux transfers carbon from the land surface to the oceans through means that are not quantified through $CO_2$ exchange at the air-surface interchange. This flux may contribute an additional 50-100 MtC yr$^{-1}$. Finally, methane, NMVOCs, and their intermediate oxidation products can be transferred directly to the oceans. As with river and aeolian transport, the subsequent fate of these products after they reach the oceans does not change their important role as a carbon-transfer
mechanism, and, therefore, these fluxes should be included. Here, we have provided a first global estimate of the size of these combined fluxes of about 50 MtC yr$^{-1}$.

The sizes of these various fluxes have been estimated in previous publications that have focused on one process or another, or they have been calculated here based on existing underlying information where no prior global estimates could be found. The novel contribution of the present analysis is bringing these fluxes together in a combined
assessment (Fig. 7), which has not previously been done. While the exact magnitude of some of these fluxes remains uncertain, it is clear that they are not zero. Their exclusion from past global carbon budgets has, therefore, systematically inflated the size of the estimated ΔB. It is, therefore, warranted to include them in future budgets and move towards a better, and less biased, estimate of ΔB and the residual sink strength of the terrestrial bisophere.

11.  **Implications for biosphere models**

The residual sink is often implicitly or explicitly equated with net exchange by the biosphere, with the two flux estimates even presented on the same graph by Ciais et al. (2013), and Le Quere et al. (2018) have referred to the residual sink as the 'land sink'. The size of the residual sink has thus been used as an important reality check of the structure and parameterisation of existing biosphere models.



However, equating the residual sink to ΔB without accounting for these additional fluxes has led to an overestimation of ΔB with important implications for our assessment of the veracity of existing biosphere models (Fig. 8). Taking the annual flux estimates generated by the average of accepted biosphere models and the size of the originally calculated residual sink, one obtains a fairly good relationship, with estimates largely conforming to a 1:1 relationship

(Fig. 8a).

If one expresses the flux estimates of the biosphere models against the revised estimates of ΔB, the match against the 1:1 line is poor. There is a large discrepancy, with the biosphere models estimating sink activity that is about 1.0–1.5 GtC yr$^{-1}$ higher than the corresponding estimates of the revised residual sink activity (Fig. 8b). This suggests that current biosphere models systematically overestimate biospheric carbon uptake, which has important implications for

present-day overall global carbon fluxes. It suggests that biosphere models may similarly overpredict future uptake rates. If enhanced carbon uptake by the terrestrial biosphere in response to climate change, including increasing atmsopheric $CO_2$, is overestimated, it will similarly overestimate the extent by which biospheric feedbacks could negate future anthroponic greenhouse gas emissions.

## 12.  General Discussion

An understanding of the global carbon cycle is important for a full appreciation of the anthropogenic disturbance of the cycle, and to what extent that disturbance is negated, or amplified, through natural feedback processes. It is even more important for the expectation of future feedback processes (e.g. Cramer et al., 2001; Friedlingstein et al., 2006; Jones et al., 2013; Huntzinger et al., 2017). It is important to anticipate whether any current carbon uptake by the biosphere may be reversed under future climatic conditions, especially under ongoing and intensifying warming (e.g.

Kirschbaum, 2000), while plants may become less responsive to $CO_2$ as atmospheric concentrations trend towards $CO_2$ saturation (e.g. Friedlingstein et al., 2006).

The comparisons between the residual sink and biospheric net $CO_2$ uptake have been given explicitly by the IPCC (Ciais et al., 2013), and they play an important role as a reality check of global biosphere models (e.g. Arneth et al., 2017; Huntzinger et al., 2017). However, to fulfil that role, it is essential that the comparisons use comparable

numbers. It is therefore important to calculate ΔB after the various known terms listed above have been explicitly quantified and subtracted from the 'residual sink'.

To anticipate correctly whether future natural biospheric carbon exchange will add to or subtract from anthropogenic emissions, it is essential to assign sink activity to the appropriate processes. If sink activity is assumed to relate to net uptake by the biosphere, one might expect it to respond to factors such as temperature, precipitation, $CO_2$

concentration, or land management, especially to the age class of forests. If one correctly infers the sensitivity of the system to these external factors, it is possible to predict future biosphere responses.

However, the various factors identified above respond to different drivers. Aeolian fluxes, for example, are probably fairly constant from year to year, or respond to climate variability like the ENSO cycle, and fluxes related to the oxidation of methane and NMVOCs would be proportional to the underlying fluxes of methane and NMVOCs.

Storage in increasing pools of plastic, bitumen and landfills, as well as cement carbonation are clearly determined by anthropogenic factors, such as economic and technological development. Future fluxes, therefore, will not respond to future temperature or $CO_2$ concentration, but need to be assessed through assessment of socio-economic developments.



Terrestrial net carbon exchange can be further sub-divided into at least four distinct processes:

### 12.1. Growth rate changes related to forest age.

Forest growth tends to be highest in young stands and decrease as stands age (Ryan et al., 1997; Kurz and Apps, 1999). Any net forest growth can therefore be due to the rebound of forest biomass in response to prior disturbance through harvesting or natural processes such as wildfire (e.g. Stinson et al., 2011). Many of the world's forests are now being inventoried at regular intervals (Pan et al., 2011) which can be supplemented with remotely sensed information (Dong et al., 2003). Growth responses can be inferred from these changes in age-class distribution (Stinson et al., 2011).

Any net carbon uptake by forests is likely to be largely due to their disturbance history. Disturbance may be through the harvesting of established forests or the planting of new ones, or due to natural factors, such as wildfire or insect-pest outbreaks. The presence of a global net forest sink implies that the rate of new growth exceeds the loss through wood extraction and other disturbance factors. A forest sink can be caused by disturbance-related carbon losses in preceding years. Understanding forest growth under current and future conditions requires disturbance effects and age-class distributions to be combined with an assessment of biophysical growth factors (e.g. Chen et al., 2000).

### 12.2. Growth rate changes related to biophysical drivers.

In principle, growth can be enhanced by increasing $CO_2$ concentrations (Pugh et al., 2016; Hickler et al., 2015), nitrogen deposition from industrial pollution (LeBauer and Treseder, 2008), or climatic changes apart from increasing $CO_2$ concentrations, such as increasing temperatures (Reyer, 2015; Sitch et al., 2015). Most modelling work has focused on these drivers as they can most easily be generalised and predicted into the future, but their actual importance is uncertain (Arneth et al., 2017; Huntzinger et al., 2017), especially in relation to age-class effects of forests that might be the principal driver of any change in the sink-source balance of forests as discussed under the previous point. It is also likely that forests subject to nutrient limitations are less responsive to changes in other biophysical drivers (e.g. Kirschbaum et al., 1998; Norby et al., 2010; Huntzinger et al., 2017) as nutrient availability retains an over-riding importance for stand productivity.

### 12.3. Blue carbon.

It has recently been recognised that mangrove forests, seagrass beds and salt marshes, can sequester large amounts of carbon, recently termed 'blue carbon' (McLeod et al., 2011; Huxham et al., 2018). It has been estimated to constitute a global carbon sink of about 200 MtC yr$^{-1}$ (McLeod et al., 2011) or even more (Breithaupt et al., 2012). However, development of coastal habitats for human infrastructure not only prevents ongoing carbon sequestration by these ecosystems but can also lead to the release of the large carbon stocks of these systems. Overall, that may result in comparable annual carbon losses as the ongoing sequestration by intact systems (e.g. Pendleton et al., 2012; Regnier et al., 2013; Atwood et al., 2017).

### 12.4. Soil organic carbon.

There may also be changes in soil carbon that can be very difficult to detect. Globally, there are about 2,500 GtC in soil organic matter to a depth of 2 m (Batjes, 2004) so that a change by just 0.4% yr$^{-1}$ would equate to a flux of 10 GtC yr$^{-1}$ to or from the atmosphere (Minasny et al., 2017). Such a change could be readily associated with land-use changes (e.g. Guo and Gifford, 2002; Kim and Kirschbaum, 2015). They may also correspond to episodic changes



within given land uses, especially changes related to accelerated erosion under agricultural land use (e.g. van Oost et al., 2007; Quinton et al., 2010; de Rose 2013).

Observational verification of annual changes of the order of 0.4% $yr^{-1}$ is extremely difficult owing to the many important factors that may positively or negatively affect soil carbon levels under different circumstances and over

different time scales (e.g. Schipper et al., 2017). However, even such proportionately small changes could be very important in the global budget and have become the basis of the recent 4 per mille initiative (e.g. Minasny et al., 2017) which aims to promote land-use practices to increase soil carbon by that amount.

## 13. Conclusions

It is important to ensure that anthropogenic $CO_2$ emissions do not lead to changes in atmospheric $CO_2$ concentrations with dangerous consequences for nature and society. A good understanding of the global carbon budget is essential for a good assessment of current and likely future trends in carbon stocks and fluxes. However, the global carbon budget in its currently used simplified form is incomplete and, therefore, does not provide appropriate guidance on the way anthropogenic and natural processes interact to lead to the observed increases in atmospheric concentrations.

These simplifications warrant modifications to the budget to explicitly and comprehensively include other known carbon fluxes between major carbon pools. While the magnitude of these various fluxes remains uncertain, understanding of the key processes has grown over the years so that it has become appropriate for these additional fluxes to be explicitly included in future global budgets.

The greatest practical importance of that inclusion lies in the role of the global budget as a reality check for the

development and parameterisation of global biosphere models. Past omission of the various known but omitted carbon fluxes discussed here is likely to have inflated the estimated sizes of natural sink activity. To provide a truer guide for the role and magnitude of these natural fluxes, it is warranted to provide a revised and more detailed assessment of the most likely changes in biospheric carbon stocks. The Global Carbon Budget is a key analysis tool for understanding the anthropogenic effect on disturbing that budget, and such ongoing refinement is warranted and necessary.

**14. Acknowledgments**

We would like to thank Robbie Andrew, Pep Canadell, Andrew McMillan, Sara Mikaloff-Fletcher and an anonymous referee for useful contributions to the underlying concepts discussed here, particularly, on the details of some of the calculations embedded in the annual carbon budget published by the Global Carbon Project, and for providing specific comments on the manuscript. We would also like to thank Anne Austin for scientific editing.

We acknowledge funding by the NZ Government's Strategic Science Investment Fund (SSIF), the UK Met Office for use of the MetUM, and the contribution of NeSI high-performance computing facilities funded jointly by NeSI's collaborator institutions and New Zealand's MBIE's Research Infrastructure programme (https://www.nesi.org.nz).





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



**Table 1: Waste generation and estimated disposal in anaerobic landfills.**

| Product | Estimated total disposed of in anaerobic landfills (Mt yr$^{-1}$) | Dry matter (%) | Carbon fraction (%) | Carbon in long-term storage (%) | Estimated storage flux (MtC yr$^{-1}$) |
|---|---|---|---|---|---|
| Wood and engineered wood products | 67 | 89 | 48 | 98 | 28 |
| Paper / paperboard | 80 | 94 | 39 | 44 | 13 |
| Plastic | 57 | 100 | 75 | 95 | 41 |
| Textile/rubber | 32 | 82 | 55 | 40 | 6 |
| Total | 236 | | | | **88** |

'Carbon in long-term storage' refers to the estimated proportion of waste stored permanently in anaerobic landfill sites. Total disposal estimates were derived from various sources including countries' greenhouse gas inventories for the Waste Sector, population statistics, IPCC documents (IPCC 2006, 2014), the European Atlas of Raw Materials (Prognos, 2008) and the World Bank Waste Reports (e.g. Hoornweg and Bhada-Tata, 2012). Moisture contents were obtained from Wang et al. (2015) and Ximenes et al. (2018). Carbon fractions were taken from the IPCC Good Practice Guidance (2014), and carbon-storage factors from Wang et al. (2011, 2015) and Ximenes et al. (2018). The dry matter and carbon fractions of the wood, engineered wood products and paper/paperboard were expressed as averages weighted by global market share of the various product categories (FAO, 2016). The estimates provided here are based on the most recent available information, but were themselves based on older information largely covering the period since 2000.





**Table 2: Summary of prior estimates of the main components of carbon fluxes through inland water ways.**

| | Influx | C efflux (evasion) | DOC | POC | Total organic | DIC | Total river | C depos. |
|---|---|---|---|---|---|---|---|---|
| Stallard (1998) | | | 230 | 300 | | 290 | | |
| Richey (2000) | | | | | | | 800 | |
| Schlünz and Schneider (2000) | | | | | 434 | 450 | | |
| Seitzinger et al. (2005) | | | 170 | 197 | | | | |
| Cole (2007) | 1900 | 750 | | | | 450 | | 230 |
| Tranvik et al. (2009) | 2900 | 1400 | | | | | | 600 |
| Romankevich et al. (2009) | | | 210 | 370 | 627[*] | | | |
| Aufdendampe et al. (2011) | 2700 | 1200 | | | | | 900 | 600 |
| Raymond et al. (2013) | | 2120 | | | | | | |
| Regnier et al. (2013) | | 1100 | 200 | 200 | | 400 | | 600 |
| Drake et al. (2018) | 5100 | 3900 | | | | | | |
| Our estimate | 2900 | 1400 | 200 | 250 | 450 | 450 | 900 | 600 |

[*]For the total organic C flux to the ocean, in addition to DOC and POC fluxes, Romankevich et al. (2009) also estimated fluxes of 25 MtC yr$^{-1}$ from coastal erosion, 14 MtC yr$^{-1}$ from ground-water influx, and 8 MtC yr$^{-1}$ from glacial run-off.



**Table 3: Estimated annual dry and wet deposition of VOCs and their oxidation products to the oceans and globally (values in brackets). Data have been calculated with the NIWA-UKCA CCM model. Units are in MtC yr$^{-1}$**

| | Global dry ocean deposition | Global wet ocean deposition |
|---|---|---|
| Formaldehyde (HCHO) | 3.4 (9.5) | 11.1 (23.1) |
| Methyl hydroperoxide (CH$_3$OOH) | 3.6 (5.1) | 5.2 (7.3) |
| Methanol (CH$_3$OH) | 2.4 (11.8) | 2.7 (6.7) |
| Formic acid (HCOOH) | 0.1 (1.2) | 2.3 (9.7) |
| Peracetic acid (CH$_3$COOOH) | 0.1 (0.8) | 1.1 (2.7) |
| Acetic acid (CH$_3$COOH) | 0.5 (1.5) | 8.6 (16.3) |
| Other C$_3$–C$_5$ isoprene and monoterpene oxidation products | 0.7 (8.3) | 8.1 (80.4) |
| Total (MtC yr$^{-1}$) | 10.8 (38.2) | 39.1 (146.2) |
| Wet + dry deposition (MtC yr$^{-1}$) | 49.9 (184.4) | |





**Table 4: Adjustments to the estimated change in the terrestrial biosphere (GtC yr⁻¹). The term ΔB$_{incLUC}$ refers to all realised biospheric carbon-stock changes, including those due to LUC whereas ΔB excludes LUC effects and includes only physiological and age-class effects. The 'inferred flux into the biosphere' is calculated as the residual sink minus cement carbonation.**

| | |
|---|---|
| Original residual uptake | 3.6 |
| Cement carbonation | -0.2 |
| Revised inferred flux into the biosphere | 3.4 |
| inland deposition | −0.6 |
| river transport (DOC, POC) | −0.45 |
| Flux of methane, NMVOC + intermediates | −0.05 |
| Aeolian dust transport | −0.05 |
| Harvested wood-products pool | −0.1 |
| Change in landfill pool originating from harvested-wood products | −0.05 |
| LUC | −1.3 |
| ΔB$_{incLUC}$ | **0.8** |
| ΔB | **2.1** |





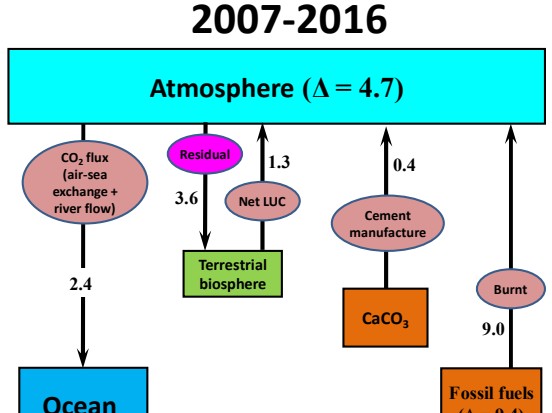

**Figure 1: The main components of the global carbon cycle for the 2007–2016 period (after Le Querre et al., 2018). Annually, 9.4 GtC yr⁻¹ of fossil fuels were used of which 0.4 GtC yr⁻¹ were not oxidised but used for manufacturing secondary products, like plastics, or incompletely combusted so that only 9.0 GtC yr⁻¹ were released to the atmosphere. The ocean flux consists of estimated air-ocean $CO_2$ exchange plus river flux of inorganic $CO_2$.**



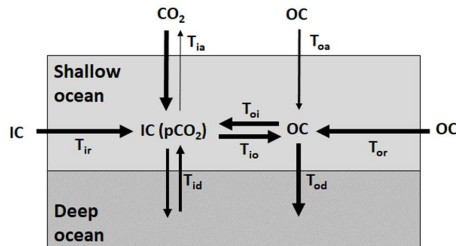

**Figure 2: Illustration of the key carbon fluxes from the atmosphere to the deep oceans, with subscripts 'i' and 'o' referring to inorganic and organic forms of carbon, respectively. $T_{ia}$ and $T_{oa}$ are exchanges with the atmosphere, $T_{ir}$ and $T_{or}$ are river**

5  **transport, $T_{io}$ and $T_{oi}$ are the interconversions between organic and inorganic forms in the ocean, and $T_{id}$ and $T_{od}$ are the transfers from shallow to deep oceans.**





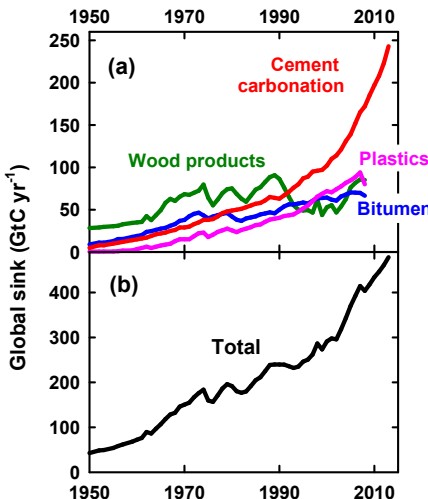

**Figure 3: Estimated net fluxes of carbon into the pools of harvested-wood products, plastics, bitumen and cement carbonation since 1950 (a) and their combined total (b). Redrawn from data given in Lauk et al. (2012) and Xi et al. (2016).**



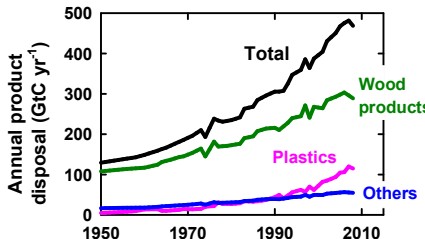

**Figure 4: Annual rates of disposal of harvested-wood products, plastics and other carbon containing compounds. Redrawn from data given in Lauk et al. (2012).**



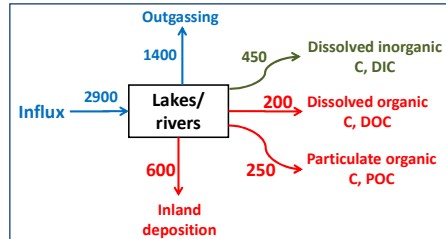

**Figure 5: The main carbon fluxes in MtC yr⁻¹ involving inland-water systems. The number shown in green is already included in the global carbon budget, whereas the number in red are added to the revised global carbon budget. The numbers in blue do not need to be included explicitly.**




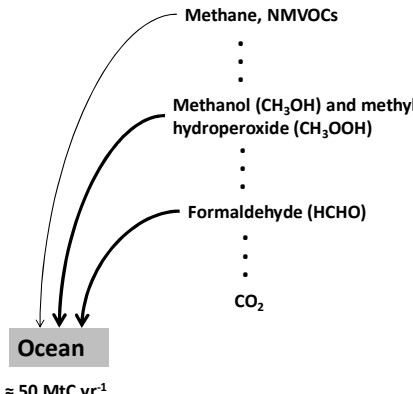

**Figure 6: The main fluxes involved in the transfers of methane and NMVOCs to the oceans. Details of the estimated overall flux are given in Table 3.**





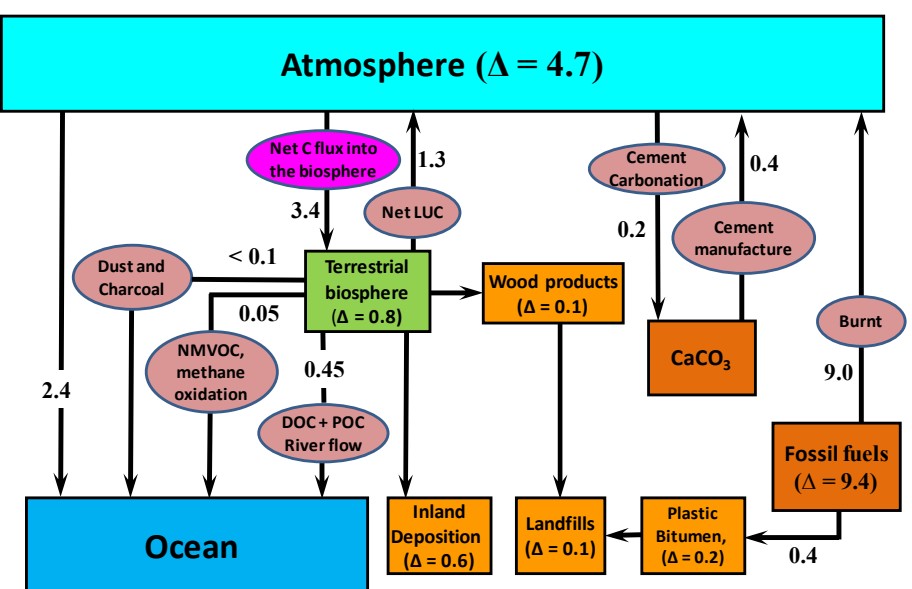

Figure 7: Expanded summary of the main components of the global carbon cycle for the 2007–2016 period. The fluxes are those given by Le Quere et al. (2018) as shown in Figure 1 above. These broad fluxes have then been modified based on
5    Table 4 and the details provided in specific sections above. Rectangular boxes refer to identified important carbon storage pools in the global carbon budget. Fluxes described in ovals refer to key fluxes between these storage pools.





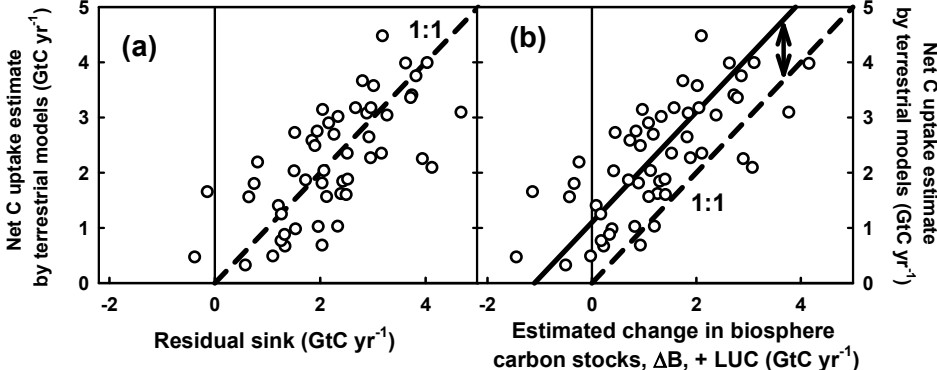

**Figure 8:** Mean estimates of net carbon uptake by the biosphere plotted against the residual sink (a), or as a function of the revised ΔB calculated here (b). Data have been taken from Le Quere et al. (2018), with each point corresponding to an
5    annual flux estimate since 1959. Data were calculated as given in Eq. 1. The dashed lines are 1:1 lines and the solid line in (b) is off-set by 1.1 GtC yr⁻¹ but retains a slope of 1.