# Peer review of "Towards a more complete quantification of the global carbon cycle"

_Biogeosciences, 2018_

## Referee Comment (RC1) · Anonymous Referee #1 · 27 Nov 2018

This is a novel and interesting paper that should stimulate discussion around this important topic. It brings together a quantification of many relatively small elements of the global carbon cycle that when combined could make a substantial reduction in the "residual sink" that has typically been assigned to the terrestrial biosphere. The paper further makes explicit some aspects which had previously been implicit in the budget – as the authors state, this improves clarity. Overall, the implications for vegetation modelling could indeed be substantial as there is an implication that current global vegetation models (which simulate an imbalance within the uncertainty of the residual sink; Le Quéré et al., 2018) may be overestimating the carbon sink provided by the biosphere (but see comment below about how this discussion is presented).

Many of the estimates included have been published elsewhere or are novel contri-

[Figure]

butions but very provisional. This paper will certainly not be the last word on those numbers, however the important thing that this paper does is to bring them all together in a consistent format and set them in the context of the global carbon budget. Careful attention has been paid to whether the fluxes considered are omitted or considered implicitly in the Global Carbon Budget as presented by Le Quéré et al. (2018). I recommend publication subject to addressing the concerns below.

Major comment

My only substantial concern relates to Section 12. The results presented by Kirschbaum et al. potentially tie in with very active discussion over the extent to which $CO_2$ fertilisation of leaf photosynthesis is propagated through to ecosystem-level increases in carbon storage (e.g. Körner, 2017, 2006; Luo et al., 2004; Medlyn et al., 2015). It is relevant to mention this however I find Section 12 generally a step too far. For instance, in section 12.1 it is stated "any carbon uptake by forests is likely to be largely due to their disturbance history". This is a valid and highly-relevant hypothesis, but it is only a hypothesis. We currently do not know the relative contributions of $CO_2$ fertilisation versus forest demography with any certainty. This should be reflected in the discussion.

Similarly, Arneth et al. (2017) is cited relating to the importance of biophysical drivers (pg. 12, line 19), but a key conclusion of Arneth et al. is that because the land-use and management change emissions may be systematically underestimated in the budget, this implies that the terrestrial "residual" sink may have previously been underestimated. Thus, it may be that the calculations presented by Kirschbaum et al. do not imply an overestimation of the carbon sink in global vegetation models, but instead account for a missing portion of the budget that balances previously underestimated land-use and management change emissions. This possibility should be explicitly laid out. Finally, the soil organic carbon section (12.4) is extremely speculative and doesn't really fit in the framework of the manuscript. Yes, a change of 0.4

Overall, in my opinion this section needs to be much more balanced, laying out the various competing hypotheses, so as to reflect a review, rather than an opinion piece.

Minor comments

Pg. 1, line 38. "net additions"? "the oceans overall are"?

Pg. 2, line 5. The budget is based on terrestrial biosphere models (TBMs) run offline, not Earth System Models.

Pg. 4, line 11. Ro or Rd?

Pg. 5, para 2. Wood product pools are included in many, if not all, of the TBMs used in Le Quéré et al. (2018). Stocks have rarely been published, which unfortunately does not facilitate a comparison, but this flux has not entirely been neglected. This should be recognised in the text.

Pg. 7, line 8. "some extra inputs mineral weathering" – does not seem to make sense. Please rephrase.

Pg. 7, line 10. Cole et al. 2009 or 2007 (cf. Table 2)?

Pg. 7, line 18/19. Repetition of material from two paragraph previously.

Table 2 is not the easiest to follow. Use of vertical lines for grouping into sections and bold text to highlight the values being carried forward would help readability.

References

Arneth, A., Sitch, S., Pongratz, J., Stocker, B.D., Ciais, P., Poulter, B., Bayer, A.D., Bondeau, A., Calle, L., Chini, L.P., Gasser, T., Fader, M., Friedlingstein, P., Kato, E., Li, W., Lindeskog, M., Nabel, J.E.M.S., Pugh, T.A.M., Robertson, E., Viovy, N., Yue, C., Zaehle, S., 2017. land-use changes are possibly larger than assumed. Nat. Geosci. 10, 79.84. https://doi.org/10.1038/ngeo2882

Körner, C., 2017. A matter of tree longevity. Science. 355, 130–131.

[Figure]

Körner, C., 2006. Plant CO2 responses: an issue of definition, time and resource supply. New Phytol. 172, 393–411.

Luo, Y., Su, B.O., Currie, W.S., Dukes, J.S., Finzi, A., Hartwig, U., Hungate, B., Murtrie, R.E.M.C., Oren, R.A.M., Parton, W.J., Pataki, D.E., Shaw, M.R., Zak, D.R., Field, C.B., 2004. Progressive Nitrogen Limitation of Ecosystem Responses to Rising Atmospheric Carbon Dioxide. BioSci 54, 731–739.

Medlyn, B.E., Zaehle, S., De Kauwe, M.G., Walker, A.P., Dietze, M.C., Hanson, P.J., Hickler, T., Jain, A.K., Luo, Y., Parton, W., Prentice, I.C., Thornton, P.E., Wang, S., Wang, Y., Weng, E., Iversen, C.M., Mccarthy, H.R., Warren, J.M., Oren, R., Norby, R.J., 2015. Using ecosystem experiments to improve vegetation models. Nat. Clim. Chang. 5, 528–534.

Quéré, C. Le, Andrew, R.M., Friedlingstein, P., Sitch, S., Pongratz, J., Manning, A.C., Korsbakken, J.I., Peters, G.P., Canadell, J.G., Jackson, R.B., Boden, T.A., Tans, P.P., Andrews, O.D., Arora, V.K., Bakker, D.C.E., 2018. Global Carbon Budget 2017. Earth Syst. Sci. Data 10, 405–448.

---

## Referee Comment (RC2) · Anonymous Referee #2 · 18 Dec 2018

The manuscript by Kirschbaum and others is a well-written summary of existing estimates of small C fluxes that should not be excluded from global C syntheses, as the authors demonstrate. I feel that it is publishable after the authors consider a number of minor points for clarity and a few more major revisions regarding deposition pathways. Namely, some dry and wet deposition terms are attributed to a flux to the ocean but in reality go to both land and ocean. In a few instances the authors appeared to be overly critical of existing budgets without justification in my opinion. The paper would also very strongly benefit from a table of abbreviations (especially equation 1!). Figure 1 is nice but doesn't link pools and fluxes with the abbreviations used in the text.

In section 2, 'The shallow ocean is too small for significant carbon storage, but the deep ocean has a huge carbon-storage capacity' seems inconsistent with the goal of

the paper to quantify small C fluxes

'As these organisms are eaten by larger organisms' is true, but small organisms also die. Regarding 'However, we believe that a more explicit representation of this pool would be desirable for greater transparency.' Yes, everyone does, but writing it as such doesn't make it clear if this will be addressed in the paper.

'However, under anaerobic conditions, breakdown effectively ceases completely' and 'never breaks down' are slight elaborations. Over meaningful time scales to the contemporary climate system perhaps. (See also Table 1 'permanently'. Readers with a long view of time may disagree.)

Wording can be simplified in many places. For example, 'Forbes et al. (2006) estimated this flux to be only small at less than 10 MtC yr–1. Could lose 'only small at'.

'any transfers to the ocean' in section 9 could also be transfers to land to the extent that NMVOCs create aerosols and cloud condensation nucei that are subsequently deposited to the surface at some point. Later in the section dry deposition (can also be wet deposition) is mentioned. This needs to be integrated more strongly with the material above. Figure 7 also needs to be modified; dust, NMVOCs, charcoal and the like also land on land.

Section 12.1 for some reason dismisses a large body of literature demonstrating that 'older' forests can take up substantial amounts of carbon, e.g. https://www.nature.com/articles/nature07276.

This sentence is an overly-harsh critique of the hard work that goes into global carbon budgeting: However, the global carbon budget in its currently used simplified form is incomplete and, therefore, does not provide appropriate guidance on the way anthropogenic and natural processes interact to lead to the observed increases in atmospheric concentrations.

Table 2: waterway is one word.

Simultaneous red and green should be avoided in Figure 5.

Figure 6 is somewhat underwhelming.

---

## Author Comment (AC1) · 15 Jan 2019

**Author Response to Interactive Comment* on "Towards a more complete quantification of the global carbon cycle" by Kirschbaum et al.**

Miko Kirschbaum and co-authors

*Correspondence to*: Miko UF Kirschbaum (KirschbaumM@LandcareResearch.co.nz)

**Response to Reviewer #1**

*Reviewer comment: This is a novel and interesting paper that should stimulate discussion around this important topic. It brings together a quantification of many relatively small elements of the global carbon cycle that when combined could make a substantial reduction in the "residual sink" that has typically been assigned to the terrestrial biosphere. The paper further makes explicit some aspects which had previously been implicit in the budget – as the authors state, this improves clarity. Overall, the implications for vegetation modelling could indeed be substantial as there is an implication that current global vegetation models (which simulate an imbalance within the uncertainty of the residual sink; Le Quéré et al., 2018) may be overestimating the carbon sink provided by the biosphere (but see comment below about how this discussion is presented).*

**Response: We would like to thank the reviewer for this positive overall assessment. The comment clearly summarises what we had intended to do with this paper.**

*Reviewer comment: Many of the estimates included have been published elsewhere or are novel contributions but very provisional. This paper will certainly not be the last word on those numbers, however the important thing that this paper does is to bring them all together in a consistent format and set them in the context of the global carbon budget. Careful attention has been paid to whether the fluxes considered are omitted or considered implicitly in the Global Carbon Budget as presented by Le Quéré et al. (2018). I recommend publication subject to addressing the concerns below*

**Response: Again, we thank the reviewer for this positive overall assessment. We also acknowledge that our paper will not be the last word on these numbers. For that reason, we have entitled it '*Towards a more complete quantification of the global carbon cycle*'. Global carbon budgets are continually evolving to reflect changing real-world fluxes, advancing scientific understanding, and the conceptual terms used to summarise observed or inferred fluxes into quantities that are deemed to be relevant to the scientific and policy-making community. Our paper aims to contribute towards that process of continual improvement.**

*Reviewer comment: Major comment My only substantial concern relates to Section 12. The results presented by Kirschbaum et al. potentially tie in with very active discussion over the extent to which CO2 fertilisation of leaf photosynthesis is propagated through to ecosystem-level increases in carbon storage (e.g. Körner, 2017, 2006; Luo et al., 2004; Medlyn et al., 2015). It is relevant to mention this however I find Section 12 generally a step too far. For instance, in section 12.1 it is stated "any carbon*

*uptake by forests is likely to be largely due to their disturbance history". This is a valid and highly-relevant hypothesis, but it is only a hypothesis. We currently do not know the relative contributions of CO2 fertilisation versus forest demography with any certainty. This should be reflected in the discussion.*

**Response:** **It had not been our intent to provide a conclusion on that ongoing debate about the various contributing**

5    **factors. The specific statement in question that '***any carbon uptake by forests is likely to be largely due to their disturbance history***' was meant to primarily refer to the pattern in individual stands for which the normal growth cycle presumably over-rides any other growth-promoting factors. We had not intended it to be seen as directly applicable to global forest carbon balances.**

**We have therefore changed that section now primarily by removing that offending sentence. We have also further**

10    **restructured that section with some additional minor wording changes. We hope this rectifies the concern expressed by the reviewer.**

***Reviewer comment:*** *Similarly, Arneth et al. (2017) is cited relating to the importance of biophysical drivers (pg. 12, line 19), but a key conclusion of Arneth et al. is that because the landuse and management change emissions may be systematically underestimated in the budget, this implies that the terrestrial "residual" sink may have previously been underestimated. Thus,*

15    *it may be that the calculations presented by Kirschbaum et al. do not imply an overestimation of the carbon sink in global vegetation models, but instead account for a missing portion of the budget that balances previously underestimated land-use and management change emissions. This possibility should be explicitly laid out.*

**Response:** **To capture the point made by Arneth et al. (2017), we have added an extra sentence to Section 12.1: '***subtler disturbance related effects on woody biomass are difficult to capture fully at the global scale and may have led to past**

20    ***underestimation of land-use change related carbon emissions (Arneth et al. 2017)....'***

***Reviewer comment:*** *Finally, the soil organic carbon section (12.4) is extremely speculative and doesn't really fit in the framework of the manuscript. Yes, a change of 0.4*

**Response:** **The reviewer's comment ended abruptly, and we are not sure what (s)he intended to say to complete the review point. At the same time, we agree with the reviewer of the speculative nature of this Section, but that is precisely**

25    **the point it was trying to make. Changes in soil carbon constitute the largest unknown contribution in the global budget. We may be able to improve the quantification of various flux by 100 MtC yr$^{-1}$ or so, but at the same time, soil carbon may change by 1 GtC yr$^{-1}$ in one direction or another without anyone being able to quantify it. We need to remain conscious of the uncertainty in our budget estimates when soil-carbon changes alone have such a large level of uncertainty. We, therefore, believe that this is an important section of the paper and have retained it.**

30    ***Reviewer comment:*** *Overall, in my opinion this section needs to be much more balanced, laying out the various competing hypotheses, so as to reflect a review, rather than an opinion piece.*

**Response:** **We are unsure what 'section' the reviewer is referring to here. If the reviewer is referring to Section 12.4, we see little 'opinion' in that section as we merely point out the existing uncertainty. If the reviewer refers to the sum-**

total of Sections 12.1 to 12.4, we aimed to do exactly what the reviewer has asked us to do: we very briefly summarised the main fluxes that could contribute to an enhanced global terrestrial sink. We tried to avoid any conclusive statement as to our view of the contributing components but simply summarised the existing literature. We are unsure what else the reviewer might want us to do to those sections.

5 *Minor comments*

*Reviewer comment: Pg. 1, line 38. "net additions"? "the oceans overall are"?*

**Response: Changes made as suggested.**

*Reviewer comment: Pg. 2, line 5. The budget is based on terrestrial biosphere models (TBMs) run offline, not Earth System Models.*

10 **Response: Change made as suggested.**

*Reviewer comment: Pg. 4, line 11. Ro or Rd?*

**Response: It should have been $R_d$. Change made to correct that.**

*Reviewer comment: Pg. 5, para 2. Wood product pools are included in many, if not all, of the TBMs used in Le Quéré et al. (2018). Stocks have rarely been published, which unfortunately does not facilitate a comparison, but this flux has not entirely*

15 *been neglected. This should be recognised in the text.*

**Response: We were aware of that inclusion of wood products in past budgets and referred to it in the original text on three separate occasions:**

**Page 2, line 34: *This flux [wood products] has already been included in net land-use change calculations (Le Quere et al., 2018), …***

20 **Page 4, line 30: *Le Quere et al. (2018) included a simple term in the calculations of net land-use change that accounted for harvested-wood products.***

**Page 9, lines 8-11: *For greater transparency, it would also be desirable to explicitly include harvested-wood products and landfill pools. The associated carbon flux is already included under the net-land-use calculations (Le Quere et al., 2018). Inclusion of a harvested-wood-products pool, therefore, would not affect the size of the residual sink, but it would require***

25 ***a corresponding adjustment of the net land-use-change flux.***

**We believe that three mentions of that inclusion of wood products in prior budgets is adequate, if not excessive already, and believe it would not be warranted to refer to its inclusion yet another time.**

*Reviewer comment: Pg. 7, line 8. "some extra inputs mineral weathering" – does not seem to make sense. Please rephrase.*

**Response: This sentence needed an extra '*from*' to say '*some extra inputs from mineral weathering*'. That has now**

30 **been corrected.**

*Reviewer comment: Pg. 7, line 10. Cole et al. 2009 or 2007 (cf. Table 2)?*

**Response: Thank you for spotting that inconsistency. It should have read '*2007*' in all references to '*Cole*'. That has now been corrected.**

*Reviewer comment:* *Pg. 7, line 18/19. Repetition of material from two paragraph previously.*

**Response:** **This partial repetition stems from the initial mention in a context where it simply listed all river related fluxes and storage items, while the second mention relates it to the fluxes and quantities that are relevant to the global carbon budget. We, therefore, regard some repetition as appropriate because the contexts are slightly different. However, we have shortened both sections to reduce the extent of that repetition.**

*Reviewer comment:* *Table 2 is not the easiest to follow. Use of vertical lines for grouping into sections and bold text to highlight the values being carried forward would help readability.*

**Response:** **To improve an understanding of the flow and grouping of the table, we have bolded our resultant estimate to indicate the numbers being carried forward. We have also omitted some of the vertical lines so that the retained vertical lines now indicate the logical grouping of some of the values.**

---

## Author Comment (AC2) · 15 Jan 2019

**Author Response to Interactive Comment* on "Towards a more complete quantification of the global carbon cycle" by Kirschbaum et al.**

Miko Kirschbaum and co-authors

5  *Correspondence to*: Miko UF Kirschbaum (KirschbaumM@LandcareResearch.co.nz)

**Response to Reviewer #2**

*Reviewer comment: The manuscript by Kirschbaum and others is a well-written summary of existing estimates of small C fluxes that should not be excluded from global C syntheses, as the authors demonstrate. I feel that it is publishable after the authors consider a number of minor points for clarity and a few more major revisions regarding deposition pathways. Namely,*

10  *some dry and wet deposition terms are attributed to a flux to the ocean but in reality go to both land and ocean. In a few instances the authors appeared to be overly critical of existing budgets without justification in my opinion.*

**Response: We like to thank the reviewer for the favourable overall assessment of the manuscript. We have addressed the specific points of criticisms in our itemised responses below.**

*Reviewer comment: The paper would also very strongly benefit from a table of abbreviations (especially equation 1!).*

15  **Response: We have general sympathy with the notion that it can often be useful to provide easy access to abbreviations used in any paper. However, in this specific paper, almost all abbreviations are used only once – for specific equations – and then immediately described in the text adjacent to the respective equations. The only abbreviations used on more than one occasion were $\Delta B$ and $\Delta B_{incLUC}$. We have renamed these symbols now into $\Delta B_{phys}$ and $\Delta B_{act}$ to stand for biomass changes due to physiological factors and actual changes, respectively. We regard these**

20  **terms as much more intuitive. We expect that with this change, it is no longer warranted to include an additional table of abbreviations.**

*Reviewer comment: Figure 1 is nice but doesn't link pools and fluxes with the abbreviations used in the text.*

**Response: We are unsure how to respond to that comment. Figure 1 gives the very abbreviated form of the global carbon cycle, with only five identified fluxes (fossil fuels, cement manufacture, land use change, ocean uptake and**

25  **residual terrestrial uptake). The main purpose of our paper was to present a more differentiated picture and add additional pools and fluxes to the budget. So, the very essence of our paper is that these additional pools are not included in the simplified version of the global budget. Showing that difference is the essence of our paper. Thus, there is little correspondence between the pools and fluxes in Figure 1, and our more complete list of pools and fluxes. It would thus be impossible to do what the reviewer is asking us to do.**

*Reviewer comment:* In section 2, 'The shallow ocean is too small for significant carbon storage, but the deep ocean has a huge carbon-storage capacity' seems inconsistent with the goal of the paper to quantify small C fluxes

**Response:** We see no inconsistency between there being a 'huge capacity', yet there being only a relatively small annual flux into that reservoir. The relatively small annual flux is still large and important in relation to the anthropogenic disturbance of the system even though it is small relative to the potential magnitude of carbon storage in the deep ocean. It just means that the flux into the reservoir has virtually no feedback effect on subsequent fluxes into that reservoir. The flux, is instead controlled by other factors. As we see no inconsistency between these statements, we have made no changes to the text.

*Reviewer comment:* 'As these organisms are eaten by larger organisms' is true, but small organisms also die.

**Response:** We have modified that statement to include the possible extra carbon fate.

*Reviewer comment:* Regarding 'However, we believe that a more explicit representation of this pool would be desirable for greater transparency.' Yes, everyone does, but writing it as such doesn't make it clear if this will be addressed in the paper.

**Response:** We have an extra part to this sentence to make it clear that such quantification is part of the present paper.

*Reviewer comment:* 'However, under anaerobic conditions, breakdown effectively ceases completely' and 'never breaks down' are slight elaborations. Over meaningful time scales to the contemporary climate system perhaps. (See also Table 1 'permanently'. Readers with a long view of time may disagree.)

**Response:** The text tries to make that assessment within the context of the contemporary carbon cycle, which is the relevant focus of the present paper. The statement is not meant to refer to a geological context. We have therefore modified that sentence to indicate that permanence refers only to a time frame relevant for carbon management.

*Reviewer comment:* Wording can be simplified in many places. For example, 'Forbes et al. (2006) estimated this flux to be only small at less than 10 MtC yr–1. Could lose 'only small at'.

**Response:** We have worked through the text once more and further tightened and simplified any text where appropriate.

*Reviewer comment:* 'any transfers to the ocean' in section 9 could also be transfers to land to the extent that NMVOCs create aerosols and cloud condensation nucei that are subsequently deposited to the surface at some point. Later in the section dry deposition (can also be wet deposition) is mentioned. This needs to be integrated more strongly with the material above. Figure 7 also needs to be modified; dust, NMVOCs, charcoal and the like also land on land.

**Response:** We are well aware of the facts mentioned by the reviewer, and these factors have been properly included in our analysis:

- **The dust deposition used in our calculations based on the work of Mahowald et al. (2005) specifically refers to dust transfer from land to oceans.**

- **The estimate of charcoal transfer of Forbes et al. (2006) specifically referred to charcoal transfer to the oceans**
- **For the transfer of NMVOC-derived compounds, we explicitly estimated the proportional deposition over land vs the oceans. This is described in detail in Supplementary Materials: "For species that are subject to dry and wet deposition, we partitioned the ocean flux as follows. We used the modelled global distribution of dry deposition fluxes to the Earth's surface of each species and accounted for deposition to the ocean using the model's land-sea mask information. Total dry deposition to the land and the ocean were then calculated by integrating the respective fluxes over the land and the ocean. In the model version used here, wet deposition fluxes were output as zonally averaged 2-dimensional fields. Therefore, we needed to partition the global wet deposition fluxes to the ocean using 3-dimensional global distributions of the species and weighted them by the global distribution of total precipitation rates."**

**So, we fully agree with the reviewer's position by the reviewer and are well aware of the importance of separating ocean and land deposition, and we believe that throughout the paper, we have used the appropriate data sources for estimating fluxes to the oceans.**

*Reviewer comment: Section 12.1 for some reason dismisses a large body of literature demonstrating that 'older' forests can take up substantial amounts of carbon, e.g. https://www.nature.com/articles/nature07276.*

**Response:** **Section 12.1 stated: '*Forest growth tends to be highest in young stands and decreases as stands age*'. That position is well-supported by the general forestry literature. Even the nature article by Luyssaert et al. (2008) cited by the reviewer agreed with that statement and showed that net ecosystem productivity of younger stands was about twice as high as that of older stands and trended towards carbon neutrality for the oldest stands in their data set.**

**At the same time, we also agree with the reviewer's point that 'older forests can take up substantial amounts of carbon'. We make no statement that would contradict that position. We simply state that younger forests can have a higher net ecosystem productivity than older forests.**

**So, our statement is well supported by a large body of forestry literature, including Ryan et al. (1997) and Kurz and Apps (1999) that have been cited in our paper, and are not contradicted by the Luyssaert et al. (2008) paper referred to by the reviewer. We, therefore, believe that this criticism is not justified, and the reviewer criticises statements that are not actually made anywhere in our paper. That makes it difficult to know how we could respond to that criticism.**

*Reviewer comment: This sentence is an overly-harsh critique of the hard work that goes into global carbon budgeting: However, the global carbon budget in its currently used simplified form is incomplete and, therefore, does not provide appropriate guidance on the way anthropogenic and natural processes interact to lead to the observed increases in atmospheric concentrations.*

**Response:** **We do not mean to be harshly critical of the global carbon budget. We recognise that overall, it provides timely and relevant information on the key carbon fluxes. Nonetheless, as we are trying to point out in our manuscripts, there are additional tweaks through inclusion of additional fluxes that would make the budget even more accurate, and that this tweak would have important consequences for our overall understanding of the current role of the biosphere,**

**in particular. So, we think the essence of our statement is correct, but we have reworded it to make it sound less critical of the valuable ongoing work on the global carbon budget.**

*Reviewer comment: Table 2: waterway is one word.*

**Response: That has now been corrected throughout the manuscript**

5 *Reviewer comment: Simultaneous red and green should be avoided in Figure 5.*

**Response: We have redrawn the figure to avoid that colour conflict.**

*Reviewer comment: Figure 6 is somewhat underwhelming.*

**Response: Our paper is trying to communicate with a wide range of potential readers with varying levels of background knowledge. Figure 6 probably contains little information for the expert reader, but we do believe that for**
10 **a reader with less expert knowledge, the figure provides a useful short summary of the relevant fluxes and the main compounds contributing to those fluxes. We have, therefore, retained that figure.**

**References**

Barlaz, M.A.: Forest products decomposition in municipal solid waste landfills. Waste Manage., 26, 321–333, 2006.

Forbes, M.S., Raison, R.J., and Skjemstad, J.O.: Formation, transformation and transport of black carbon (charcoal) in
15 terrestrial and aquatic ecosystems. Sci. Total Environ., 370, 190–206, 2006.

Kurz, W.A. and Apps, M.J.: A 70-year retrospective analysis of carbon fluxes in the Canadian forest sector. Ecol. Appl., 9, 526–547, 1999.

Lauk, C., Haberl, H., Erb, K.H., Gingrich, S., and Krausmann, F.: Global socioeconomic carbon stocks in long-lived products 1900–2008. Environ. Res. Lett., 7, Article Number: 034023, 2012.

20 Sebastiaan Luyssaert, S., Schulze, E.-D., Börner, A., Knohl, A., Hessenmöller, D., Law, B.E., Ciais, P. and Grace, J. (2008). Old-growth forests as global carbon sinks. Nature, 455, 213–215.

Mahowald, N.M., Baker, A.R., Bergametti, G., Brooks, N., Duce, R.A., Jickells, T.D., Kubilay, N., Prospero, J.M., and Tegen, I: Atmospheric global dust cycle and iron inputs to the ocean. Global Biogeochem. Cycles, 19, Article Number: GB4025, 2005.

Ryan, M.G., Binkley, D., and Fownes, J.H.: Age-related decline in forest productivity: Pattern and process. Adv. Ecol. Res.,
25 27, 213–262, 1997.